# Hemorrhagic Cerebral Insults and Secondary Takotsubo Syndrome: Findings in a Novel In Vitro Model Using Human Blood Samples

**DOI:** 10.3390/ijms231911557

**Published:** 2022-09-30

**Authors:** Serge C. Thal, Manuel Smetak, Kentaro Hayashi, Carola Y. Förster

**Affiliations:** 1Department of Anesthesiology, Helios University Hospital Wuppertal, University Witten/Herdecke, 42283 Wuppertal, Germany; 2Emergency and Pain Medicine, Intensive Care, Department of Anesthesiology, University of Würzburg, 97070 Würzburg, Germany; 3Advanced Stroke Center, Shimane University Hospital, Izumo 693-8501, Japan

**Keywords:** Takotsubo syndrome, subarachnoid hemorrhage, inflammation, in vitro, blood, blood–brain barrier, human, patient, endothelial cells

## Abstract

Intracranial hemorrhage results in devastating forms of cerebral damage. Frequently, these results also present with cardiac dysfunction ranging from ECG changes to Takotsubo syndrome (TTS). This suggests that intracranial bleeding due to subarachnoid hemorrhage (SAH) disrupts the neuro–cardiac axis leading to neurogenic stress cardiomyopathy (NSC) of different degrees. Following this notion, SAH and secondary TTS could be directly linked, thus contributing to poor outcomes. We set out to test if blood circulation is the driver of the brain–heart axis by investigating serum samples of TTS patients. We present a novel in vitro model combining SAH and secondary TTS to mimic the effects of blood or serum, respectively, on blood–brain barrier (BBB) integrity using in vitro monolayers of an established murine model. We consistently demonstrated decreased monolayer integrity and confirmed reduced Claudin-5 and Occludin levels by RT-qPCR and Western blot and morphological reorganization of actin filaments in endothelial cells. Both tight junction proteins show a time-dependent reduction. Our findings highlight a faster and more prominent disintegration of BBB in the presence of TTS and support the importance of the bloodstream as a causal link between intracerebral bleeding and cardiac dysfunction. This may represent potential targets for future therapeutic inventions in SAH and TTS.

## 1. Introduction

Spontaneous subarachnoid hemorrhage (SAH) is the key manifestation of hemorrhagic stroke and is caused primarily by the rupture of an intracranial aneurysm [1,2]. While the bleeding and elevated intracranial pressure cause brain injury, clinical outcome in patients surviving the initial hemorrhagic event is thought to be largely determined by brain edema formation, cerebral hypoperfusion, and large vessel vasospasm [3,4,5,6,7,8], which occur days to weeks following the initial event [6].

Vascular disruption is the underlying cause of cerebral hemorrhage extending from intracerebral, subarachnoid to intraventricular hemorrhage [9]. The disease ontogeny culminates in cerebral hemorrhage-induced blood–brain barrier (BBB) disruption, which contributes an imperative component to brain injury after the initial cerebral hemorrhagic event. BBB dysfunction, in turn, drives vasogenic edema, permits leukocyte extravasation, and may open the passage of potentially neurotoxic and vasoactive compounds into the brain parenchyma.

In the acute stage of SAH, electrocardiogram (ECG) alterations are observed in 50–100% of patients. In some severe cases, even left ventricular regional wall-motion dysfunction, a characteristic of Takotsubo syndrome (TTS), is often documented [10]. The presentation of TTS is more frequent in females, suggesting a prominent role of gender in the neuro–cardiac axis [11]. The severity of aneurysmal SAH, as determined by the Hunt and Hess scale [12], appears to be significantly associated with the development of secondary TTS. Vice versa, cardiomyopathies can also present with neurological manifestations [13].

SAH disrupts the neuro–cardiac axis and leads to neurogenic stress cardiomyopathy (NSC) of variable severity (reviewed in [14,15]). Several studies have shown that decreased cardiac output (CO) can induce both acute [16,17] and chronic cerebral hypoperfusion [18,19] and that impaired left ventricular (LV) function is associated with cognitive decline [20,21]. We recently demonstrated that cerebral perfusion correlated significantly with left ventricular end-diastolic volume after murine SAH [22].

Pathophysiological triggers may be due to neurohormonal activation and catecholamine surges; cardiac manifestations are commonly detailed in the setting of SAH [23]. It is largely accepted that severe acute brain injuries cause cardiac dysfunction ranging from ECG changes to Takotsubo syndrome (TTS) [24]: TTS, better known as the transient left ventricular apical ballooning or broken-heart syndrome, is a recently described acute cardiac syndrome [25,26,27] and characterized by a transient left ventricular apical akinesis and hyperkinesis of the basal left ventricular portions. As of late, interestingly, even prior non-diagnosed varieties of the degree of dysfunction were presented with a hypokinetic midportion of the left ventricle and a hyperkinetic apex [27]. Despite the acute onset, TTS patients do not show obstructive atherosclerotic coronary disease. In turn, cardiac akinesis remains a life-threatening condition that may worsen cerebral perfusion in the presence of cerebral vasospasm.

This report presents a novel in vitro model of SAH mimicking the effects of blood or serum exposure, respectively, on BBB integrity in vitro using monolayers of murine brain microvascular endothelial cells (cerebEND) [28] derived from the cerebellum as an established model of the BBB. We specifically considered the vessel-damaging effects of plasma concentration of biomarkers reflecting the neuroendocrine response in TTS (catecholamines, certain pro-inflammatory cytokines) [29,30,31]. We provide data on changes in brain endothelial junction proteins subjected to blood (with/without clot-related factors), allowing further insights into the mechanisms underlying junction modification.

## 2. Results

To present an example of the general presentation of subarachnoid hemorrhage (SAH), Figure 1 displays a native CT image of a 72-year-old female aneurysmal SAH patient on day 0. The ruptured aneurysm of the arteria communicans anterior is indicated with a black arrow. The basal cisterns are filled with blood (hyperdense material, blue arrow). In addition, the gyri are blurred, which is most likely a result of (reactive) brain edema. The aneurysm was occluded by coil embolization on the same day resulting in complete remission of the neurological deficits and enabling her to return to her usual life.

In order to simulate subarachnoid hemorrhage under in vitro conditions, Transwell cell culture systems were used. These systems are a side-by-side vertical diffusion system that creates an upper and lower compartment separated only by a microporous membrane. The upper compartment represents the vascular side, as cerebEND cells are seeded onto the membrane as part of the BBB. Accordingly, the lower compartment is the parenchymal side of the brain. Until the cerebEND reached confluence, medium was added to both compartments to prevent the cells from drying out. So as to simulate subarachnoid hemorrhage, the medium in the lower compartment was replaced by human serum to facilitate contact with the medium from above and the serum from below. Since the compartments are separated by a microporous membrane, the diffusion of small molecules (e.g., from the serum) is possible.

In order to investigate time-dependent changes in protein expression, differentiation medium was added to the upper compartment, whereas serum of healthy postmenopausal patients was added to the lower compartment, followed by incubation for various periods of time (30 min, 1 h, 2 h, 4 h and 8 h). Changes in protein levels were determined by Western blot analysis and compared to the 30-min incubation as a control group (Figure 2). Claudin-5 expression was significantly reduced after 1 h (0.61 ± 0.048 x-fold). After 2 h, the protein levels were further reduced (0.55 ± 0.052 x-fold). The steady decline was also present after 4 h (0.47 ± 0.061 x-fold) and 8 h (0.31 ± 0.003 x-fold). Occludin expression also showed a steady decrease, but no significant reduction was present after 1 h incubation. A significant decrease in occludin protein level was measured after 2 h (0.58 ± 0.053 x-fold), 4 h (0.59 ± 0.094 x-fold), and 8 h (0.43 ± 0.009 x-fold).

So as to investigate the change in protein expression, differentiation medium was placed in the upper compartment, whereas serum from healthy patients (postmenopausal and premenopausal patients) and serum from patients with various heart diseases (acute ACS, acute TTS, SAH with secondary TTS) in the lower compartment and incubated for 4 h. Changes in protein levels were determined by Western blot analysis and compared to the healthy postmenopausal patients as a control group (Figure 3). The groups were also compared with one another. Compared to the incubation with healthy postmenopausal serum, claudin-5 was significantly reduced after incubation with healthy premenopausal patients (0.75 ± 0.036 x-fold), acute TTS serum (0.81 ± 0.035 x-fold) and SAH with secondary TTS (0.51 ± 0 x-fold). There was also a significant reduction of claudin-5 after incubation with serum from acute TTS patients and SAH with secondary TTS compared to patients with acute ACS (1.21 ± 0.099 x-fold). The incubation with serum from acute ACS patients had no significant effect on the expression of claudin-5 compared to the incubation with serum from healthy postmenopausal patients. In contrast to that, healthy premenopausal serum showed a significant reduction compared to acute ACS (0.75 ± 0.036 x-fold).

The other targets (occludin, ZO-1 and VE-cadherin) showed no significant changes in the protein level (Table 1).

Since actin is important for the stabilization and maintenance of the blood–brain barrier, actin-phalloidin staining of cerebEND cells were performed to visualize changes after incubation with serum from patients with various heart diseases. Actin filaments look similar in both healthy patient populations (Figure 4a,b). The cells retained their homogenous endothelial monolayer and the typical elongated spindle-shaped morphology. After incubation with acute TTS serum (Figure 4c), the cell morphology changed slightly with less fine elongated and spindle-shaped morphology. The actin filaments also looked damaged and less fine. In addition, the nuclei looked dulled compared to the nuclei after incubation with postmenopausal serum. After incubation with subacute TTS serum (Figure 4d), the typical cell monolayer was still visible but not as fine and detailed in comparison to the other images. The actin filaments look more damaged and lacerated since there are blacker empty areas, and the filaments are less fine compared to the other patients. In addition, the nuclei after incubation with subacute TTS serum look even more dulled and indistinct compared to the acute TTS patient.

## 3. Discussion

Aneurysmatic subarachnoid hemorrhage (SAH) is the major type of hemorrhagic stroke. The hemorrhagic event itself is fatal in about 30% of cases. Most surviving patients consistently present brain damage [3,4,5] but also cardiac dysfunction. This is a first report on an in intro model employing SAH patient blood samples to establish the link between cardiac dysfunction and BBB dysfunction after SAH.

### 3.1. SAH In Vitro Modeling

Vascular disruption is the main cause of cerebral hemorrhage, including intracerebral, subarachnoid and intraventricular hemorrhagic events [9]. SAH has been shown to induce disruption of blood–brain barrier (BBB) integrity, contributing to brain injury after the initial cerebral hemorrhage [9]. BBB dysfunction drives, e.g., vasogenic edema formation, enables leukocyte extravasation and may permit entry of potentially neurotoxic and vasoactive xeno- and endobiotics into the brain. Persisting vascular injury, a failure to restore integrity at damaged sites, or the occurrence of hemorrhage in additional vascular sections cause further hematoma expansion. Despite the high clinical relevance of cerebral hemorrhagic events, the underlying mechanisms and exact sequence of events remain poorly understood. In contrast to studies addressing ischemic stroke events, this shortage may mainly be attributed to the lack of suitable in vitro models which would allow for the variation and investigation of specific molecular factors and individual cell types independently of one another [32,33].

In this study, we present a novel in vitro model of SAH mimicking the effects of full blood or serum on BBB integrity using in vitro monolayers of murine brain microvascular endothelial cells (cerebEND) [28] derived from the cerebellum as an established model of the BBB (Figure 1). We studied the time dependency of blood/serum exposure on BCECs forming the BBB by monitoring the expression of key tight junction proteins Claudin-5, occludin, ZO-1 and the adherens junction protein VE-cadherin. We consistently demonstrate a time-dependent pattern of the protein expression in cerebEND cells after incubation with blood serum from healthy individuals with protein content lowering effects on the tight junction proteins claudin-5, occludin, ZO-1 and the adherens junction protein VE-cadherin (Figure 2, Table 1).

Endothelium blood or serum exposure from the basolateral (brain parenchymal) side entails a common mechanism that initiates BBB breakdown. Several events that may lead to brain endothelial barrier disruption have been reported, including alterations in expression or localization of proteins forming the apical junctional complex (tight junctions TJ, adherens junctions, AJ), linking damage to endothelial cells to endothelial cell death and increased endothelial cell transcytosis [9]. Our study highlights the time dependency, targets and impact of barrier impairment and its control by blood-borne factors on BBB disintegration in the setting of SAH. This is, to our knowledge, the first in vitro pathology model of SAH concentrating on the effects of blood or serum exposure on brain capillary endothelial cell integrity, as assessed by evaluating effects on the TJ components Claudin-5, occluding, ZO-1 and VE-cadherin, the adherens junction protein (Figure 2, Table 1); we demonstrate that exposure to both, patient full blood or serum, respectively impairs the endothelial barrier integrity. This observation not only closes the scientific gap between animal models and human disease but also strengthens the idea of barrier impairment as a common underlying pathological principle.

In SAH, BBB disruption has been reported as early as 10 minutes after ictus and can persist for up to seven days (reviewed in Tso and Macdonald [34]). A biphasic opening has been reported with a peak between 3 and 72 h in the rat endovascular perforation model [35]. In SAH patients, enhanced BBB permeability on MRI imaging predicts the development of delayed cerebral ischemia [36], a major contributor to poor outcomes in patients. SAH induces a marked neuroinflammatory response in both animals and patients, including an influx of leukocytes into the brain [37].

Several therapeutic approaches have shown efficacy in reducing BBB disruption and junction changes after SAH in preclinical models [9]. It should be noted, however, that it remains uncertain to date whether these effects were directly on the cerebral endothelium or indirectly by reducing parenchymal injury. Due to this persisting lack of knowledge, agents that specifically target the endothelium (and BBB structure) had not been examined.

Our study may now help to disseminate such potential targets for future therapies in SAH, neuroinflammation and TTS cases. This is important to the advent of SAH, and this is also a common predictor of adverse outcomes in patients [38].

### 3.2. The Actin Cytoskeleton in SAH

The actin cytoskeleton is a major regulator of cellular stiffness, alterations in its structure influence the cellular mechanical phenotype, which in turn can be linked to distinct cellular function, thus being a potential diagnostic marker [9]. A variety of conditions result in the production of actin stress fibers [9] and those fibers may exert forces on junction proteins causing junction disruption and barrier opening as observed more rapidly and pronounced in the presence of TTS acute phase and subacute phase serum exposure.

In conjunction with this, we show that the incubation with serum from patients resulted in a pronounced reduction of protein levels of the endothelial adherens junction protein, VE-cadherin. The reduction of VE-cadherin was strongest when incubated with serum from patients diagnosed TTS acute and TTS acute in comorbidity with SAH.

Our experiments indicate that the perturbation of BBB integrity by blood or serum exposure influences actin dynamics, cellular morphology and ultimately BBB integrity. Interestingly, increasing the duration of blood exposure or content in barrier-compromising pro-inflammatory cytokines and catecholamines promoted the reduction in TJ protein expression and re-organization of actin filaments [39].

Besides the barrier-forming proteins in the apical junctional complex of brain capillary endothelial cells, the cellular cytoskeleton is a key contributor to BBB formation and the function and positioning of TJ within the apical junction complexes [40]. The cytoskeleton provides a scaffold for junction location and transmits forces that open junctions (e.g., during stress fiber formation). The cytoskeleton is moreover essential for vesicular trafficking, and endocytosis is the chief aspect of junction remodeling [9]. The molecular regulation of actin cytoskeletal organization has been investigated mainly in inflammatory and ischemic disorders [40,41], while its role in SAH remains to be examined.

This is despite evidence that actin cytoskeletal reorganizations (e.g., stress fiber formation) or the reduction in actin contents are a therapeutic target for reducing BBB disruption in cerebral ischemia [42], and ischemia is a major component of SAH-induced injury [43]. Cytoskeletal changes have been examined in smooth muscle cells after SAH [44], and it will be important to separate the importance of changes in both cell types. We show in this report for the first time that serum exposure remodeled the actin cytoskeleton and reduced the total amounts of actin in endothelial cells comprising the BBB. In the samples co-incubated with TTS patient serum, even a reduction in the total amount of actin was observed (Figure 4).

TJs and AJs within the apical junctional complex are linked to the cytoskeleton, so the observed alterations in actin orientation and content might be partly explained by the observed reduction and function of VE-cadherin (Figure 3, Table 1), which has been described to be mediating intercellular adhesion together with the catenins, its cytoplasmic plaque proteins [41].

As described for the above-mentioned conditions, underlying the observed events, several kinases (e.g., RhoA, ROCK, Src, JNK, Agrin as a mediating component of extracellular matrix between endothelial cells and astrocytes at the BBB) and alterations in cAMP levels may be involved in junction changes and BBB disruption after cerebral hemorrhage [9]. In that context, Fasudil, a ROCK inhibitor, has been extensively tested in SAH, but with a focus on vasospasm [45]. In addition, there is evidence that ROCK inhibition reduces BBB damage and TJ changes in animal models of cerebral ischemia [46].

Limitations of our study shall also be mentioned: We only use endothelial cells as a monolayer, but the blood–brain barrier is more complex as, for example, pericytes and astrocytes and components of the extracellular matrix have important roles in maintaining the barrier as well. To completely understand processes regarding the blood–brain barrier, it is crucial to take all aspects of the BBB into account. As indicated in the above description, there are moreover multiple pathways involved that can lead to alterations in TJ function and BBB disruption after cerebral hemorrhage. The importance of those pathways may vary with time, so the different phases of BBB disruption will have to be followed up more closely in matching in vitro and in vitro experiments in the future.

### 3.3. Cardiac Dysfunction following SAH

As ample evidence has been given for an association of TTS with catecholamines and mediators of inflammation [30], fueling microvascular endothelial dysfunction and critical vascular conditions in regions of the central autonomic network important for cardiac function [47], we set out to investigate the effects of blood from patients presenting with ACS, TTS and TTS-SAH to determine differential effects on the timing and extent of BBB impairment. While, the brain–heart interaction within TTS has been appreciated with changes in the ANS and limbic system [48,49], we now give direct proof for enhanced alterations in barrier properties of brain microvascular endothelial cells in vitro following exposure to TTS patient blood containing increased catecholamine levels and characteristic inflammatory mediators potentially driving the development of SAH or hemorrhagic events. According to our observations reported in this study, direct barrier-compromising effect of these substances fueling SAH and hemorrhagic events in comorbidity with TTS can be concluded (Table 1, Figure 3).

About 40–100% of SAH patients show pathological electrocardiogram (ECG) readings and sonographic signs of heart failure. The situation progresses to developing life-threatening cardiac symptoms in an estimated 5% of these patients [15,50]. Recent evidence showed that even patients without prior history of cardiac disorders are affected, implicating a causal link between SAH and cardiac symptoms [15,51,52]. Population-based analysis of the SAH population further revealed that the female gender is a key risk factor for SAH and cardiac dysfunction such as TTS [53,54].

Even though there has been a growing knowledge of the importance of early brain injury [34,55], studies on the time course of effects on the distinct brain endothelial junction proteins are not available to date. Little is also known about the causal link between SAH and other extracerebral organ dysfunctions, such as neurogenic lung edema [10].

For the relation between TTS-stress factors and hemorrhagic events in vitro, we were able to point out a speedier and more articulated deterioration of brain endothelial monolayer integrity in the presence of patient blood from cases presenting with ACS, TTS and secondary TTS-SAH in comorbidity [24] as opposed to blood from ACS patients or healthy subjects, respectively, suitable to be considered as an initiating event driving the development of SAH in the context of secondary TTS, culminating in TTS-related heart failure as a manifest form of end-organ damage [30].

Specifically, cerebrovascular damage in the insula cortex resulting from SAH or hemorrhagic stroke might, in this context, lead to structural cerebrovascular damage, potentially leading to sympatho-vagal imbalance [26,47] by the resulting disturbances in CNS homeostasis [56]. One possible explanation for this might be the release of a combination of factors stimulating local neurons and leading to the activation of downstream sympathetic pathways resulting in cardiac events such as TTS [57]. In using the experimental hemorrhagic stroke model in rats, recent investigation reports autonomic and cardiovascular consequences resulting from performing the hemorrhagic stroke model. Specifically, damage to the insula cortex was reported to result in serious cardiovascular consequences and even lateralization of the characteristics [57].

Clinically, recent reports recommend assessing patients elected for mechanical thrombectomy [58] for acute ischemic stroke for Takotsubo cardiomyopathy even in the case of successful recanalization, given that the neuroanatomical network of the insular cortex seems to play a critical role in autonomic control of cardiac activity [59].

### 3.4. Limitations of Our Study

The available in vitro BBB models have specific limitations and are, therefore, not perfect model systems. The development of in vitro BBB models has been driven by the need for a fast, reliable and cost-effective tool and to reduce complexities (both structural and functional) of the BBB as well as for screening of putative CNS drugs. Specifically, for the investigation of complex insults such as stroke, brain trauma or SAH, the paramount role of astrocytes and other cell types comprising the neurovascular unit need to be considered. Our study is therefore limited, and future approaches need to include astrocytes and pericytes in a 3D model, or at least the use of conditioned medium [58].

One general limitation of in vitro BBB models is the lack of physiological aspects of the neurovascular unit (NVU). One key component in living organs is, e.g., sheer stress, which is similar between arteries and capillaries and markedly lower in venules. The present study used a static BBB model to focus on paracellular barrier properties and intracellular signaling. Our findings, therefore, do not incorporate the physiological effects of blood flow and sheer stress, which require a dynamic in vitro (DIV) model of the BBB with an intraluminal flow through artificial capillary-like structural supports.

## 4. Materials and Methods

### 4.1. Cell Culture

Cerebellar endothelial (cerebEND) cells were isolated, immortalized and cultivated, as described previously [28]. For the simulation of SAH in in vitro conditions, cerebEND cells were grown to confluence for five days using Transwell systems. Therefore, the cells were placed on 6-well or 12-well Nunc Polycarbonate Cell Culture Inserts (Thermo Fisher Scientific, #140640 or #140652, Waltham, MA, USA) with pore sizes of 0.4 µm coated with gelatin. Dulbecco’s Modified Eagle’s Medium (DMEM)-high glucose (Sigma-Aldrich, #D5796, St. Louis, MI, USA) with the addition of 10% heat inactivated-fetal calf serum (FCS) and 50 U/mL penicillin-streptomycin was used as growth medium. The medium was placed in the lower and upper compartments of the Transwell system. After growing to confluence, differentiation was induced for 24 h using DMEM-high glucose with 1% FCS and 50 U/mL penicillin-streptomycin. The medium in the lower compartment was then removed and serum was added, causing the cells to have contact with differentiation medium from above and serum from below (only divided by the microporous membrane). The added serum was either from patients without any major cardiovascular or endocrinological disease (further called healthy patients) or from patients with various heart diseases. The healthy patients had no altered coronary arteries in the coronary angiography diagnostics, and only minor cardiovascular or endocrinological diseases were accepted (e.g., arterial hypertension, diabetes mellitus). The patients with various heart diseases were either patients with Takotsubo syndrome (acute, subacute or chronic phase) or patients with acute coronary syndrome (ACS).

### 4.2. Human Patient Serum

Patient serum preparation from human blood samples was performed, as described previously [32]. We sampled blood from 28 pseudonymized patients and subsequently divided these into 5 different groups: postmenopausal healthy patients, *n* = 6, premenopausal healthy patients, *n* = 6, ACS acute (<6 weeks), *n* = 2, TTS acute (<6 weeks), *n* = 13, TTS acute + SAH (<6 weeks), *n* = 1. As healthy controls, subjects were selected that did not present with altered coronary arteries in the coronary angiography diagnostics, while minor cardiovascular or endocrinological diseases were accepted (e.g., arterial hypertension, diabetes mellitus). All patients were diagnosed according to the InterTAK Diagnostic Criteria [33].

For serum preparation, all blood samples were drawn from subjects using S-Monovette collection tubes (Sarstedt) and incubated at room temperature for 60 min. Then, the blood samples were centrifuged at 1500× *g* for 15 min, and the serum was isolated, immediately frozen and stored at −80 °C until analysis. All the methods used in this study were performed in accordance with the relevant guidelines and regulations.

### 4.3. Ethics Approval and Consent to Participate

The study protocol was approved by the Hiroshima City Asa Hospital Research Committee (01-3-3) as previously described [34], Hiroshima, Japan and was conducted in accordance with the principles stated in the Declaration of Helsinki. All participants provided informed written consent. From July 2019 to September 2019, hospitalized TTS patients were admitted consecutively at Hiroshima City Asa Hospital. All subjects included in this analysis had a good prognosis. None of the patients died during hospitalization. In the Kruskal–Wallis test, there were no significant differences with respect to age (median age 83 vs. 80 yrs, *p* = 0.48) and sex (78% vs. 100% female, *p* = 0.23) between the group with TTS and control.

### 4.4. Western Blot

Cells were washed with ice-cold PBS (twice) and then lysed on ice with ice-cold RIPA Buffer containing a protease inhibitor cocktail (Roche). After the harvest, the cells were sonicated (10 times for 0.5 s and 20 W) and centrifuged (10 min, 4 °C, 11.000 rcf). The supernatant was saved, and the protein content was quantified with Pierce BCA Protein Assay Kit (Thermo Fisher Scientific, # 23225). Then, 20 µg protein (mixed 1:4 with 4 × Laemmli containing 6% β-mercaptoethanol) was separated by electrophoresis using self-made SDS-polyacrylamide electrophoresis gels (separation gel 8%), and subsequently transferred overnight at 4 °C on a PVDF membrane. These membranes were blocked for 1 h with 5% non-fat dry milk in PBS and incubated with the primary antibody (overnight at 4 °C) in 1% BSA in PBS. After washing with 0.1% Tween in PBS (PBS-T, 3 times for 10 min) and blocking with 5% non-fat dry milk in PBS (20 min), the membranes were incubated with the secondary antibody in 1% BSA in PBS (1 h). As the primary antibody we used: mouse anti-claudin-5 (1:1000, Thermo Fisher Scientific, #35-2500), guinea pig anti-occludin (1:100, Acris, #358-504), rabbit anti-ZO-1 (1:1000, Thermo Fisher Scientific, #61-7300), anti-β-actin-HRP (1:25000, Sigma-Aldrich, #A3854), anti-tubulin antibody [YL1/2] (1:10000, abcam, #ab197740), anti-goat VE-cadherin (1:200, Santa Cruz Biotechnology, #sc-6458, Dallas, TX, USA). As secondary antibodies, we used: horse anti-mouse IgG (1:3000, Cell Signaling Technology, #7076S, Danvers, MA, USA), goat anti-guineapig IgG (1:5000, Santa Cruz Biotechnology, #sc2438), goat anti-rabbit IgG HRP-linked (1:3000, Cell Signaling Technology, #7074S), mouse anti-goat IgG-HRP (1:3000, Santa Cruz Biotechnology, sc-2354). For imaging, the membranes were washed with PBS-T (3 times for 10 min) and incubated in ECL solution (2 min). The images were taken with FluorChem FC2 Multi-imager II (Alpha Innotech). The ImageJ software was used to determine the density of the protein bands.

### 4.5. Real-Time qPCR

For RT-PCR, the cells were washed twice with sterile PBS. The cells were then harvested, lysed and the RNA isolated and purified according to the manufacturers’ instructions for Nucleospin RNA (Macherey-Nagel, #740955). A total of 1 µg of the RNA was converted to cDNA using a High-Capacity cDNA Reverse Transcription Kit (Thermo Fisher Scientific, #4368814) in a 2720 Thermal Cycler (Thermo Fisher Scientific, # 4359659). The Real-Time qPCR was performed using TaqMan Fast Advanced Mastermix (Thermo Fisher Scientific, #4444557) along with the synthesized cDNA and the following TaqMan Gene Expression Assays (Thermo Fisher Scientific): claudin-5 (Mm00727012_s1, #4331182), occludin (Mm00500912_m1, # 4331182) and calnexin as endogenous control (Mm00500330_m1, # 4331182). The measurements were performed in a StepOnePlus Real-Time PCR System (Thermo Fisher Scientific, #4376600).

### 4.6. Actin-Phalloidin Staining

For actin-phalloidin staining, cells were incubated with serum in Transwell systems, as described above. Cells were prepared according to the manufacturers’ instruction of Acti-stain 488 (Cytoskeleton, #PHDG1). Briefly, the cells were washed with PBS once, followed by incubation in 3.7% paraformaldehyde in PBS for 10 min. After washing in PBS for 30 s, cells were incubated in 0.1% triton-x-100 in PBS for 5 min. After washing twice for 30 s in PBS, cells were incubated in 5% pig serum in PBS for 1 h and afterward washed 3 times for 30 s in PBS. After the preparation, staining was performed with 100 nM of Acti-stain 488 and 100 nM DAPI. Coverslips were fixed onto slides by using Mountant permaflour (Thermo Fisher. #TA-030-FM). Afterward, analysis was performed with a Keyence fluorescence microscope (BIOREVO BZ-9000).

### 4.7. Statistical Analysis

Statistical analysis was performed using GraphPad Prism 9. Data are presented as mean ± standard error of the mean. The number of independent experiments is indicated in or under the figures. Statistical significance was evaluated by one-way ANOVA with either Tukey’s correction for multiple comparisons or with Dunnett’s multiple comparisons test. Statistical significance was assumed for *p* < 0.05.

## 5. Conclusions

This is the first description of an in vitro model of SAH and alterations in TJ and AJ proteins as a functional pathway leading to BBB disintegration following serum exposure; moreover, we have identified important actin cytoskeletal reorganization and reduction of content in fibers in endothelial cells of the BBB. Together with the evidence that serum exposure of TTS patients has the most pronounced effect, targeting BBB endothelial and cytoskeletal integrity is a putative novel target for the treatment of SAH and secondary TTS.

## Figures and Tables

**Figure 1 ijms-23-11557-f001:**
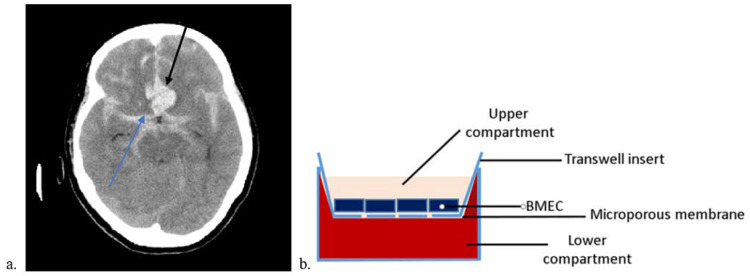
Example of a subarachnoid hemorrhage CT scan and scheme of the in vitro model. (**a**) Symmetric subarachnoid hemorrhage with aneurysm of the arteria communicans anterior; Shimane University Hospital, Izumo, Japan; (**b**) in vitro model of subarachnoid hemorrhage. The blood–brain barrier, which is formed by brain microvascular endothelial cells (BMEC), divides the system into two compartments (vascular side: upper compartment, brain parenchymal side: lower compartment).

**Figure 2 ijms-23-11557-f002:**
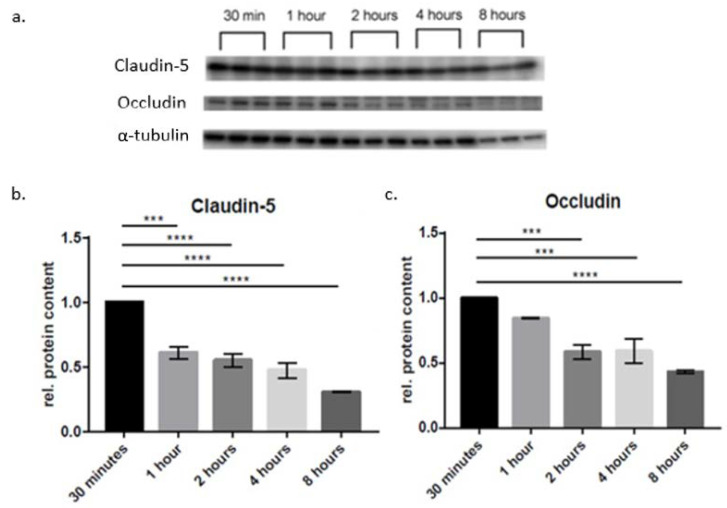
In vitro simulation of the subarachnoid hemorrhage and the time-dependent change in protein expression in cerebEND cells after incubation with serum from healthy patients. Changes of protein expression in cerebEND cells after incubation with serum for different periods of time. (**a**) Membranes of the Western blots used in this experiment. (**b**) Expression of claudin-5 after 30 min, 1 h, 2 h, 4 h and 8 h of incubation with serum. (**c**) Expression of occludin after 30 min, 1 h, 2 h, 4 h and 8 h of incubation with serum. Changes in protein expression were normalized to α-tubulin. Data are the means (±SEM) of 3 independent experiments. Statistical significance was evaluated using Dunnett’s multiple comparisons test. ***: *p* < 0.001; ****: *p* < 0.00001.

**Figure 3 ijms-23-11557-f003:**
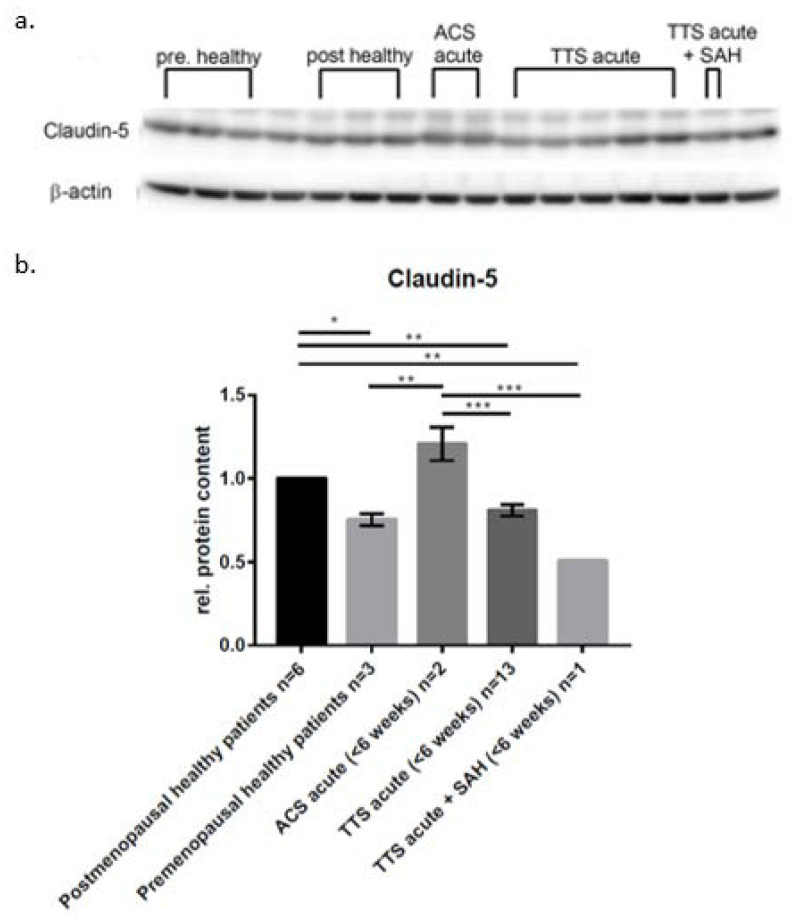
In vitro simulation of a subarachnoid hemorrhage with sera from patients with various heart diseases and its effects on protein expression in cerebEND cells. Changes in the protein expression of claudin-5 in cerebEND cells after simulating SAH with serum from patients with various heart diseases and incubation for 4 h. (**a**) Membranes of the Western blots used in this experiment. (**b**) Expression of Claudin-5 after 4 h of incubation with serum. Claudin-5 protein expression was normalized to β-actin. Data are the means (±SEM) of independent experiments. Statistical significance was evaluated using Tukey’s multiple comparisons test. *: *p* < 0.05; **: *p* < 0.01; ***: *p* < 0.001.

**Figure 4 ijms-23-11557-f004:**
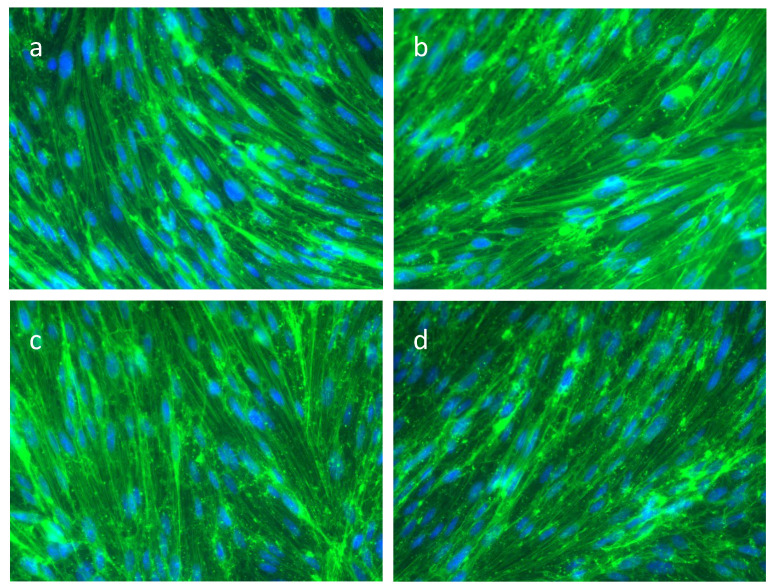
Actin-phalloidin staining of cerebEND cells after incubation with serum of patients with various heart diseases. Actin-phalloidin staining of cerebEND cells incubated for 4 h with (**a**) healthy postmenopausal serum, (**b**) healthy premenopausal serum, (**c**) acute TTS serum and (**d**) subacute TTS serum. Actin filaments: green, nuclei: blue. Images were taken at 60× magnification.

**Table 1 ijms-23-11557-t001:** Occludin, ZO-1 and VE-cadherin protein levels in cerebEND cells.

	Occludin	ZO-1	VE-Cadherin
Postmenopausal healthy patients, *n* = 6	1	1	1
Premenopausal healthy patients, *n* = 6	0.96 ± 0.156	0.82 ± 0.209	1.07 ± 0.281
ACS acute (<6 weeks), *n* = 2	1.1 ± 0.083	1.12 ± 0.042	1.4 ± 0.181
TTS acute (<6 weeks), *n* = 13	0.78 ± 0.057	0.77 ± 0.041	0.69 ± 0.082
TTS acute + SAH (<6 weeks), *n* = 1	0.69	0.76	0.67

Protein levels after simulating SAH with serum from patients with various heart diseases and incubating for 4 h. Protein expression was normalized to β-actin. Data are the means (±SEM) of independent experiments. Statistical significance was evaluated using Tukey’s multiple comparisons test.

## Data Availability

The data that support the findings of this study are available from the corresponding author, C.Y.F., upon reasonable request.

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
