# Peer review of "Hemorrhagic Cerebral Insults and Secondary Takotsubo Syndrome: Findings in a Novel In Vitro Model Using Human Blood Samples"

_ijms, 2022, doi:10.3390/ijms231911557_

Round 1
Reviewer 1 Report
This excellent work points out that targeting blood-brain barrier (BBB) endothelial and cytoskeletal integrity may constitute a novel target for treatment of subarachnoid hemorrhage (SAH) and secondary Takotsubo syndrome (TTS).
In order to capture the attention of a wide readership, authors are highly encouraged to provide an original image as a visual abstract that clearly represents their hypothesis and findings described along their discussion.
Authors have used a simplified BBB in vitro model based on one cell type (cerebEND cells), the brain microvascular endothelial cells derived from mice (Silwedel, C.; Forster, C., Differential susceptibility of cerebral and cerebellar murine brain microvascular endothelial cells to loss of barrier properties in response to inflammatory stimuli. J Neuroimmunol 2006, 179, (1-2), 37-45). As the authors are fully aware, other neurovascular unit cells such as pericytes and astrocytes are known to play an important role in maintaining the barrier both in vivo and in vitro. Therefore, authors should highlight this limitation in the present study.
Author Response
Comment 1:
In order to capture the attention of a wide readership, authors are highly encouraged to provide
an original image as a visual abstract that clearly represents their hypothesis and findings
described along their discussion.
Response: Thank you very much for suggesting this presentation tool. As requested, we
provide a graphical overview on SAH/TTS and the findings of the study.
Comment 2:
Authors have used a simplified BBB in vitro model based on one cell type (cerebEND cells),
the brain microvascular endothelial cells derived from mice (Silwedel, C.; Forster, C.,
Differential susceptibility of cerebral and cerebellar murine brain microvascular endothelial
cells to loss of barrier properties in response to inflammatory stimuli. J Neuroimmunol 2006,
179, (1-2), 37-45). As the authors are fully aware, other neurovascular unit cells such as
pericytes and astrocytes are known to play an important role in maintaining the barrier both in
vivo and in vitro. Therefore, authors should highlight this limitation in the present study.
Response: The major limiting factor is the lack of availability of reliable human BBB models. In
addition, these different in vitro models are not perfect model systems and show specific
characteristics to investigate specific scientific questions. In the present study, we set out to
find ways to model the complex and challenging situation of secondary TTS.
We included a sentence/ paragraph:
The available in vitro BBB models have specific limitations and are therefore not perfect model
systems. The development of in vitro BBB models has been driven by the need for a fast,
reliable, and cost-effective tool and to reduce complexities (both structural and functional) of
the BBB as well as for screening of putative CNS drugs. Specifically, for the investigation of
complex insults such as stroke, brain trauma or SAH, the paramount role of astrocytes and
other cell types comprising the neurovascular unit need to be considered. Our study is
therefore limited and future approaches need to include astrocytes and pericytes in a 3D
model, or at least the use conditioned medium [1].
Reviewer 2 Report
ijms-1846578
Hemorrhagic cerebral insults and secondary takotsubo syndrome: Findings in a novel in-vitro model using human blood samples.
This is an interesting study of a novel in-vitro model of spontaneous subarachnoid hemorrhage, which attempts to preserve blood-brain barrier integrity, by using monolayers of murine brain microvascular endothelial cells.
Although the analysis is well planned, the authors do not clarify in respect with one of the most relevant factors involved in endothelial characteristics: the blood flow. The solid conclusion after these results should be strongly supported by the consideration of this crucial physiological component, in culture medium.
The characteristics of endothelial cells depend, at least, of the cellular features (mainly cytoskeleton), substrate tissue and blood flow. Probably, blood flow and arterial pressure determine the main behavior of endothelial cells.
How the authors introduce, control and evaluate the effects of physiological blood flow and pressure on the culture model?
Author Response
Comment 1:
Although the analysis is well planned, the authors do not clarify in respect with one of the most
relevant factors involved in endothelial characteristics: the blood flow. The solid conclusion
after these results should be strongly supported by the consideration of this crucial
physiological component, in culture medium.
The characteristics of endothelial cells depend, at least, of the cellular features (mainly
cytoskeleton), substrate tissue and blood flow. Probably, blood flow and arterial pressure
determine the main behavior of endothelial cells.
How the authors introduce, control, and evaluate the effects of physiological blood flow and
pressure on the culture model?
Response: We fully agree with the reviewer. The development of “realistic” in vitro blood-brain
barrier (BBB) models, which recapitulate physiological parameters and molecular aspect of the
neurovascular unit (NVU), is of fundamental importance not only in CNS drug discovery, but
also in translational research. Successful modeling of the NVU would provide a most valuable
tool to identify the pathological factors and mechanism of action (and targets) of CNS
disorders.
Different approaches have been developed to mimic the BBB in vitro. This includes static and
dynamic (flow-capable) platforms, as well as the use of different cell types such primary cells,
immortalized cell lines and more recently, stem cells.
On key component in living organ is sheer stress, which is similar between arteries and
capillaries and markedly lower in venules. We therefore acknowledge the importance of the
aspects raised by the reviewer, which is addressed in dynamic in vitro (DIV) model of the BBB.
These models allow the use of co-cultures and create intraluminal flow through artificial
capillary-like structural supports. Unfortunately, they have several disadvantages, e.g. these
systems are not intended to be used in HTS studies, require much more time and technical
skills to be established, and require a high cell load for the initial setup of the system. Moreover,
the designs do not allow visualization of the intraluminal compartment to assess morphological
and phenotypic changes of the endothelium. In addition, DIV models have limited availability
(due to cost constrains).
For our present study we focused on paracellular barrier properties and intracellular signaling.
Therefore, we decided to use monocultures, using Petri dishes and/ or trans well filter inserts
until the DIV models will be better developed. In our in vitro methods, BECs can be used in
large quantities for biochemical and physiological studies. Moreover, optical characteristics of
the plastic Petri dishes allow to see and locate cells easily, the cells can easily be quantified,
and finally experimental costs are markedly lower.
Literature:
1. Yamasaki, T.; Hayashi, K.; Shibata, Y.; Furuta, T.; Yamamoto, K.; Uchimura, M.; Fujiwara, Y.; Nakagawa,
F.; Kambara, M.; Yoshikane, T.; Nagai, H.; Akiyama, Y.; Tanabe, K.; Tanabe, J., Takotsubo
cardiomyopathy following mechanical thrombectomy for acute ischemic stroke: illustrative case. J
Neurosurg Case Lessons 2021, 2, (9), CASE21372.
Round 2
Reviewer 2 Report
The authors do not include the suggested experimental procedures to analyze the effect of blood flow in this study.
At least they could include the comments of their answer in the new version.
Author Response
We thank the reviewers for their critical evaluation of our manuscript and their valuable suggestions. We believe that our report has been substantially strengthened by addressing their concerns. We have revised the work and hope that the current version meets the requirements of the reviewers and is now suitable for publication in International Journal of Molecular Sciences. Please find below our point-by-point responses to the reviewers’ comments.
Reviewer 2
Comment 1:
The authors do not include the suggested experimental procedures to analyze the effect of blood flow in this study.
At least they could include the comments of their answer in the new version.
Response:
Setting up a new in-vitro model using a dynamic concept with an intraluminal flow through artificial capillary-like structural supports does take a very long time. We are very happy about the very good suggestion by the reviewer and for bringing the blood flow aspect to our attention. At present we are evaluating how to set up the experiments to continue our research in a dynamic in-vitro model for future research.
As suggested by the reviewer we have included this important limitation of in-vitro BBB research in the limitations section and marked the changed passage yellow.